# Shear Strength of Geopolymer Concrete Beams Using High Calcium Content Fly Ash in a Marine Environment

**Muhammad Sigit Darmawan [1],\***  **, Ridho Bayuaji [1], Hidajat Sugihardjo [2], Nur Ahmad Husin [1] and Raden Buyung Anugraha Affandhie [1]**

[1]  Department of Civil Infrastructure Engineering, Faculty of Vocational Studies, Institut Teknologi Sepuluh Nopember, Surabaya 60111, Indonesia; bayuaji@ce.its.ac.id (R.B.); nahusin@ce.its.ac.id (N.A.H.); r_buyung_aa@ce.its.ac.id (R.B.A.A.)

[2]  Department of Civil Engineering, Faculty of Civil, Environment, and Geo Engineering, Institut Teknologi Sepuluh Nopember, Surabaya 60111, Indonesia; hidayat@ce.its.ac.id

\*  Correspondence: msdarmawan@ce.its.ac.id; Tel.: +62-81-331-457-141

**Abstract:** This paper deals with the behavior of a geopolymer concrete beam (GCB) under shear load using high calcium content fly ash (FA). The effect of the marine environment on the shear strength of GCB was considered by curing the specimen in a sea splashing zone for 28 days. Destructive and non-destructive tests were carried out to determine the properties of geopolymer concrete in different curing environments. Geopolymer concretes cured at room temperature showed higher compressive strength, slightly lower porosity, and higher concrete resistivity than that of those cured in sea water. From the loading test of the GCB under shear load, there was no effect of a sea environment on the crack pattern and crack development of the beam. The shear strength of the GCB generally exceeded the predicted shear strength based on the American Concrete Institute (ACI) Code.

**Keywords:** Geopolymer; Fly Ash; Marine Environment; Shear Strength; Beam

## 1. Introduction

This study is part of the research entitled geopolymer precast concrete, carried out at the Material and Building Structure Laboratory, Department of Civil Infrastructure Engineering, Institut Teknologi Sepuluh Nopember Indonesia. The final goal of this research is the utilization of geopolymer concrete for precast concrete in Indonesia. Geopolymer has so far been considered as the "greener material" compared to ordinary concrete due to its utilization of waste material without any cement.

Research on geopolymer concrete is on-going to clarify its mechanical properties and durability. The mechanical properties of geopolymer concrete, including compressive strength, indirect tensile strength, modulus of elasticity, and poison ratio have been investigated [1]. The results showed that the modulus of elasticity increased with increasing compressive strength and the poison ratio was in the range of 0.12 to 0.16. The behavior of the indirect tensile strength and the range of collapse under stress conditions was very similar to that of Ordinary Portland Cement (OPC) concrete. The geopolymer concrete stress-strain curve showed a peak strain in the range 0.0024–0.0026. The mechanical properties of geopolymer concrete have been predicted using Australian Standards [2], which include the modulus of elasticity, poison ratio, compressive strength, splitting tensile strength, and flexural strength. The activator concentration and type of fly ash affected the quality of the mechanical properties of the geopolymer concrete and some of the superior mechanical properties, namely the rapid strength and small shrinkage [3]. The bond strength of geopolymer concrete was

also higher than that of OPC concrete [4]. Geopolymer concrete used low calcium fly ash showing higher compressive strength when cured under elevated temperatures [5].

The permeability character and the modulus of elasticity of the geopolymer concrete were highly dependent on the geopolymer concrete compressive strength [6]. Geopolymer concrete shrinkage was very small, around 100 micro-strains after one year. This value was smaller than the OPC concrete shrinkage of 500 to 800 micro-strains [7]. The increase in Sodium Hydroxide (NaOH) molarity leads to a decrease in chloride penetration and reduced corrosion in steel [8]. Because of that, the durability of geopolymer concrete is better than normal concrete, as it is resistant to sulfate and chloride [9] and a carbonation environment [10]. The mechanism of geopolymerization from 28 days to 356 days has been studied with results of durability increase, diffusion coefficients decrease, and sustainable gel production leading to denser microstructure [11]. Other research showed that geopolymer concrete offered better properties than ordinary concrete such as rapid strength development [12,13].

Rapid strength development is the key determining factor in the success of the precast concrete industry while durable concrete is the main solution for concrete structure design to operate in aggressive environments such as the marine environment. The behavior of the geopolymer concrete beam (GCB) under flexure was examined in a previous study [14]. In this study, fly ash (FA) content used for making GCB was 504 kg/m$^3$ and was combined with the use of NaOH solution of 14 moles (M) to achieve the design compressive strength of 25 to 30 MPa. Despite the use of this rich mix, the compressive strength achieved was only 14 MPa. The absence of heat curing in the making of the GCB was first considered as the main factor that contributed to this low compressive strength. However, after a careful examination and many trial mixes were performed to get the design compressive strength, the absence of heat curing was not found to be the main factor contributing to the low compressive strength of GCB. One possible reason for the failure to achieve the design compressive strength in a previous study was an error in the weighing process of the material used in the concrete mix.

The study of GCB under flexure has been extensively investigated by many researchers [14–20]. On the other hand, only a few studies dealt with the behavior of the GCB under shear [18,21]. Therefore, this study was performed to increase the understanding of the behavior of the GCB designed to fail under shear. The effect of chloride environment on the shear strength of the GCB was also considered by curing the specimens in a seawater environment. Earlier studies showed that the cracking load of the GCB cured in seawater environment increased up to 2.75 times than that of the GCB cured at room environment [9,14]. In addition, high calcium content fly ash (FA) was utilized as low calcium content FA is becoming difficult to find in Indonesia, bearing in mind that the setting time is still an issue to be tackled when utilizing this type of FA for wider practical applications [14,22].

## 2. Research Method

### 2.1. Specimens Details

Two series of experimental investigations (I and II) with a total of six reinforced GCB were made and then loaded to failure to determine their behavior under shear loading:

- Series I—beams cured at room temperature (B1, B2, and B3)
- Series II—beams cured in seawater at the splashing zone (B4, B5, and B6)

All the test beams were 100 mm wide and had an overall depth of 150 mm. The total length of each beam was 1500 mm. Three beams were tested for each series. Two 13 mm diameter of rebar Grade 400 were used as tension reinforcement whereas an 8 mm diameter of rebar Grade 240 with a spacing of 250 mm was used as shear reinforcement for the two series. Figure 1 shows the geometry and reinforcement details of the GCB.

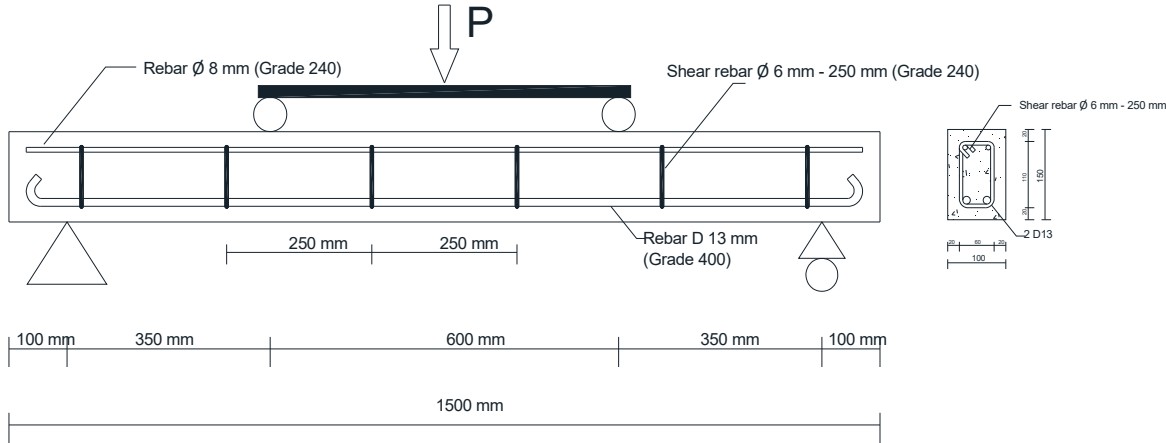

**Figure 1.** Specimen detail.

## 2.2. Materials

Table 1 summarizes the chemical composition of FA used in this study, which is determined by X-Ray Fluorescence (XRF). It shows that the Calcium oxide (CaO) content of the FA exceeds 10%. Therefore, based on the American Society for Testing and Materials (ASTM) C618 [23] this FA can be categorized as type C. The chemical composition of FA used in a previous study is also shown in Table 1 for comparison purposes. Even though both FA came from the same source (i.e., the Paiton Coal Power Plant), the chemical composition of the two FA differed considerably, especially their $Al_2O_3$, CaO, and $Fe_2O_3$ contents. This is a very common trend for waste material such as FA to have a high variation of its chemical properties.

**Table 1.** Chemical composition of FA.

| Oxides | Weight (%) | |
|---|---|---|
| | This Study | Previous Study [14] |
| $SiO_2$ | 40.60 | 41.97 |
| $Al_2O_3$ | 10.19 | 15.55 |
| CaO | 17.98 | 14.15 |
| MgO | 4.21 | 6.19 |
| $Na_2O$ | 0.77 | 2.26 |
| $K_2O$ | 1.80 | 1.73 |
| $TiO_2$ | 1.03 | 0.87 |
| $Fe_2O_3$ | 20.43 | 14.16 |
| $SO_3$ | 2.06 | 2.12 |
| $P_2O_5$ | 0.23 | 0.22 |
| MnO | 0.16 | 0.12 |
| SrO | 0.54 | 0.65 |
| Total | 100.00 | 100.00 |

Table 2 shows the concrete composition used in the present study to achieve the design compressive strength of 25 to 30 MPa, together with the composition used in a previous study for comparison purposes. It shows that the present study used 8M of NaOH whereas the previous study used 14M of NaOH and used less FA content. The sodium silicate ($Na_2SiO_3$) used in this study had the proportion of $H_2O$ 13.49%, $SiO_2$ 56.38%, and $Na_2O$ 30.13%. The reduction of molarity and FA content was made to get a longer setting time and produce a less expensive mix. Coarse and fine aggregates used in this study were very similar to the aggregates used in the previous study [14]. A commercially available superplasticizer SP425 produced by Fosroc was used to extend the setting time.

**Table 2.** Geopolymer concrete mixture.

| Material | Quantity (kg) | |
| --- | --- | --- |
| | This Study | Previous Study [14] |
| Fly ash | 468 | 504 |
| Coarse aggregate | 1008 | 1008 |
| Fine aggregate | 672 | 672 |
| NaOH | 84 (8M) | 86.4 (14M) |
| $Na_2SiO_3$ | 168 | 123.2 |
| Superplasticizer | 9.36 | 15.2 |
| Added water | 0 | 10.08 |
| $Na_2SiO_3$: NaOH | 2.0 | 1.43 |

Concrete cylinders of 100 mm × 200 mm were tested prior to the GCB loading test to determine the compressive and splitting tensile strength of geopolymer concrete. For each series and each test, three cylinders were made. These cylinders were cured simultaneously with the beam at the designated environment. In addition to compressive and splitting tensile strength tests, the following tests were also carried out on the concrete cylinders to better understand the material properties of FA-based geopolymer concrete:

- The ultrasonic pulse velocity (UPV) test
- Resistivity test
- Porosity test

The yield strength of the rebar used in GCB was determined by the tensile testing of the rebar. The average yield strength of Grade 400 and Grade 240 rebar were 500 MPa and 300 MPa, respectively. Based on this test data, the capacity of the GCB to support the load can be estimated and then compared with the actual failure load obtained from the loading test.

### 2.3. Mixing and Curing

Mixing of the geopolymer concrete was started by the dry-mixed FA and fine aggregates for at least five minutes and then followed by the addition of the coarse aggregate in stages into the dry mix. The alkaline solution and superplasticizer were then added sequentially to the dry mix and the entire batch was mixed for another five minutes. Note that the alkaline solution (NaOH and $Na_2SiO_3$) was prepared separately and the NaOH solution was prepared 24 hours before mixing.

The appearance of geopolymer concrete straightway after mixing is shown in Figure 2. The mixture was very flowable, and the measured slump was 190 mm. However, the mixture soon became very sticky and started to set within 35 minutes after the mixing stopped. Despite reducing the molarity of the NaOH from 14M to 8M in the mix and using less FA content, the change of setting time was almost insignificant (only five minutes difference). The use of superplasticizer to extend the setting time had no effect. This is not surprising as the commercially available superplasticizer was designed for normal concrete and not for geopolymer concrete. This is an area for further investigation. Figure 3 shows the beam immediately after casting.

Two days after casting, the beams and their controlled cylinders were removed from their molding and then cured in the designated environment. Three beams and six cylinders were kept at room temperature of 35 °C for the first series while the other three beams and six cylinders (the second series) were sent to the Port of Tanjung Perak Surabaya and placed in a splashing zone to experience cycles of wetting and drying due to the tide cycles, see Figure 4. Unfortunately, one of the beams (B6) was lost during the curing in the seawater, and therefore for the second series, only two beams were available for the loading test. After 28 days, all the beams were taken back to the laboratory and then loaded until failure. For the second series, the specimens were first allowed to fully dry for 2 days before loaded to failure.

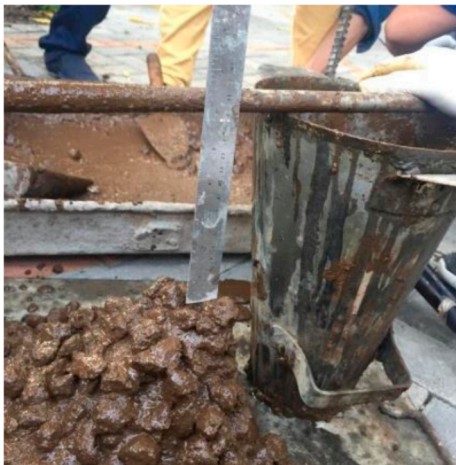

**Figure 2.** Slump test of the geopolymer concrete mix.

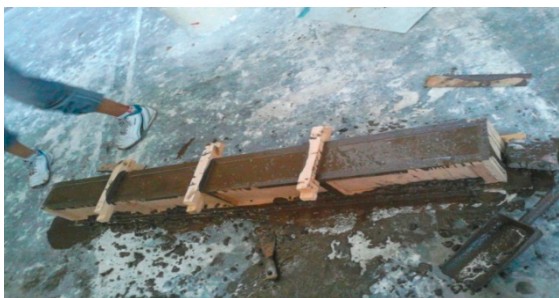

**Figure 3.** The geopolymer concrete beam after casting.

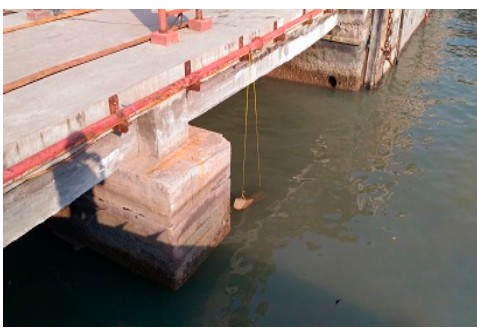

**Figure 4.** Geopolymer concrete beam cured in seawater for 28 days.

The seawater at Port of Tanjung Perak was tested to determine the concentration of its main aggressive elements that influence the degree of the chloride attack. Two samples of water were tested, and the results are presented in Table 3. The table shows that the highest chloride and sulfate content of the sea water was 19,000 mg/L and 2471 mg/L, respectively. These values were higher than the chloride and sulfate content of seawater found in the Port of Paiton, approximately 150 km away from Port of Tanjung Perak [24] of 15720 mg/L and 1778 mg/L, respectively. These higher contents are possibly caused by the higher water pollution at Port Tanjung Perak than that of Port of Paiton.

**Table 3.** Chemical content analysis of sea water.

| Parameter | Sample 1 | Sample 2 |
|:---:|:---:|:---:|
| pH | 7.70 | 7.25 |
| Chloride (mg/L) | 19,000 | 17,000 |
| Sulfate (mg/L) | 2471 | 2203 |

### 2.4. Testing Procedure

All beams were simply supported and tested over a span of 1300 mm using a hydraulic test rig with a capacity of 1000 KN. Two-point loads were applied symmetrically on the span with a distance of 300 mm from the middle of the span as shown in Figure 5. The load was applied and increased monotonically until the beam failed in the shear mode. At each increment load, the deflection was measured, and the crack formed was then examined visually and marked. The deflection of the beam due to the load was monitored by one Linear Variable Displacement Transducer (LVDT) and two dial gauges, see Figure 6. The LVDT was placed in the mid-span while the dial gauge was placed directly below the applied load. The average of the three readings from the transducer and dial gauge was recorded as the displacement of the beam. All loads and the deflection data were manually and electronically recorded. The final mode of failure was also carefully observed.

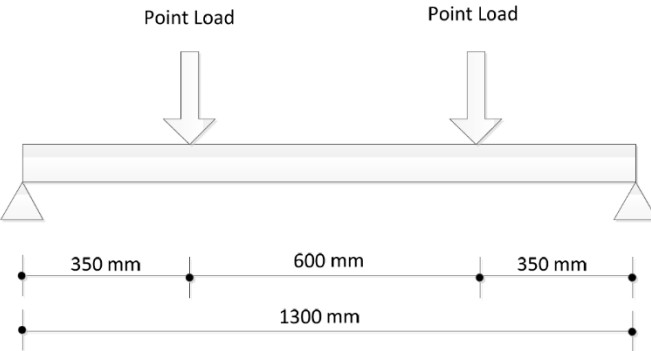

**Figure 5.** Loading test arrangement.

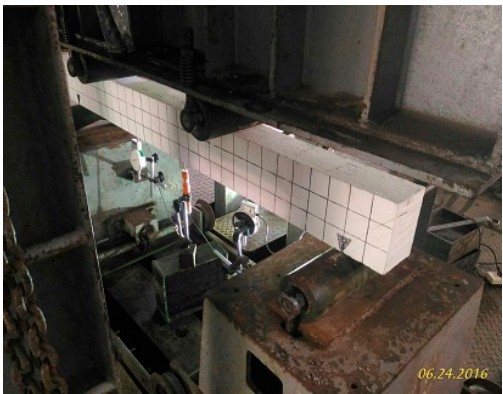

**Figure 6.** Loading test of geopolymer concrete beam (GCB).

## 3. Results and Discussion

### 3.1. Compressive and Splitting Tensile Strength of Concrete Cylinders

Table 4 summarizes the compressive and splitting tensile strength of the concrete cylinders from the series I and II tests. The individual compressive strength from the series I test was higher than the individual compressive strength of the series II test. Based on the average compressive strength, the series I was 41% higher than that of series II. This clearly shows that the temperature played a significant role in the strength development of the geopolymer concrete and not the type of environment. The room temperature used was around 34 °C whereas seawater temperature was around 25 °C. Note that the room temperature of 34 °C was obtained naturally in the laboratory without adding cost to the production of the GCB. This is an important finding as heat curing is mostly used for low-calcium FA-based geopolymer concrete to obtain the design compressive strength. In term of strength variation, the series I test had a higher strength variation than series II. As will be shown later, the test result of

the ultrasonic pulse velocity test, resistivity test, and porosity test also had a similar trend, where series I had a higher variation of material properties than series II.

**Table 4.** Compressive and splitting tensile strength of geopolymer concrete.

| Series | Sample | Compressive Strength (MPa) | Average | Splitting Tensile Strength (MPa) | Average | Ratio to Compressive Strength |
|--------|--------|---------------------------|---------|----------------------------------|---------|-------------------------------|
| I | B1 | 25.97 | | 2.38 | | |
| | B2 | 32.84 | 35.09 | 2.73 | 2.91 | 0.08 |
| | B3 | 46.47 | | 3.63 | | |
| II | B4 | 22.02 | | 2.96 | | |
| | B5 | 24.82 | 24.78 | 3.08 | 3.05 | 0.12 |
| | B6 | 27.50 | | 3.12 | | |

Table 4 also indicates that the average splitting tensile strength of the two series were very close one to another. In fact, series II has a slightly higher average splitting tensile strength than series I. Contrary to the other study [14], the average ratio between splitting and compressive strength in this study was between 0.08–0.12, which is very similar to that of normal concrete.

### 3.2. Ultrasonic Pulse Velocity (UPV) Test

The UPV test can be used to assess the uniformity and relative quality of concrete. This test was performed in accordance with ASTM C597-09 [25]. However, due to the non-destructive nature of this test, the result of this test must be accompanied by a destructive test for validation. Table 5 shows the results of the UPV test performed on series I and II specimens. It shows that based on the average of the UPV test results, specimens from series II had a slightly better quality than that of specimens from series I. This result contradicts with the result of compressive strength results which shows that series I had a higher compressive strength. Based on results from BS 1881 [26], concrete with a UPV test result between 3000–3500 m/s could be classified as medium quality concrete.

**Table 5.** Ultrasonic Pulse Velocity (UPV) test of the geopolymer concrete samples.

| Series | V (m/s) | | Average V (m/s) | Average Compressive Strength (MPa) |
|--------|---------|------|-----------------|------------------------------------|
| I | 2580 | 2637 | | |
| | 2697 | 3845 | 3087 | 35.09 |
| | 3180 | 3580 | | |
| II | 2967 | 3430 | | |
| | 3087 | 3347 | 3237 | 24.78 |
| | 3387 | 3207 | | |

### 3.3. Resistivity Test

Concrete resistivity is considered as one of the most important parameters that can help to examine corrosion of steel in concrete. Research has found that there are direct correlations between concrete resistivity and both the initiation and the propagation period [27]. Material which has higher resistivity also has higher corrosion resistance. In this study, resistivity is measured using commercially available equipment called Resipod, made by Proceq. The Resipod is designed to measure the electrical resistivity of concrete. A current is applied to the two outer probes, and the potential difference is measured between the two inner probes. The current is carried by ions in the pore liquid. Table 6 gives the resistivity of the geopolymer concrete beams for series I and II specimens. It shows that series I specimens, which had higher strength, had a higher resistivity than series II specimens. Thus, in this study concrete resistivity test result corresponded well with the compressive strength test result. However, based on the available research data [27], concrete with a resistivity of less than 5 kΩcm

is considered to experience heavy corrosion. Note that these data were mostly obtained for normal concrete and therefore for geopolymer concrete, this criterion needs further justification.

**Table 6.** Resistivity test of geopolymer concrete beams.

| Series | Resistivity (kΩcm) | | Average Resistivity (kΩcm) | Average Compressive Strength (MPa) |
|---|---|---|---|---|
| I | 3.90 | 4.93 | 3.99 | 35.09 |
|  | 3.33 | 5.10 |  |  |
|  | 3.03 | 3.67 |  |  |
| II | 2.23 | 2.30 | 2.25 | 24.78 |
|  | 2.07 | 2.23 |  |  |
|  | 2.23 | 2.43 |  |  |

### 3.4. Porosity Test

Concrete porosity affects both the strength and durability of the concrete structure. Concrete with low porosity has a high concrete strength and high durability. The porosity of concrete can be determined using the saturation method, helium pycnometry, and mercury intrusion porosimetry. In this study, the total porosity was determined using the vacuum saturation apparatus [28]. The result of this test is presented in Table 7. Table 7 indicates that geopolymer concrete cured at room temperature has a slightly lower porosity than that of geopolymer concrete cured in a seawater environment. The Table also shows that concrete with lower porosity had higher strength. However, the concrete strength difference (41%) was greater than the concrete porosity difference (4%). The data in the literature show that for concrete strength of 30 to 40 MPa, the porosity of normal concrete was usually around 15 to 20% [29,30].

**Table 7.** Porosity test of geopolymer concrete samples.

| Code | Porosity (%) | Average Porosity (%) | Average Compressive Strength (MPa) |
|---|---|---|---|
| Series I | 10.33 | 10.94 | 35.09 |
|  | 11.07 |  |  |
|  | 11.43 |  |  |
| Series II | 10.83 | 11.37 | 24.78 |
|  | 11.11 |  |  |
|  | 12.11 |  |  |

### 3.5. Crack Development and Mode of Failure

Flexural (vertical) cracks first developed in the region of the higher moment (at mid-span between the two-point loads). With increasing load, additional bending cracks formed throughout the shear span. At the same time, the existing cracks lengthened and widened. In the shear span region, as the cracks grew longer, the vertical cracks developed into inclined cracks due to the shear action. After application of approximately 70% of the ultimate load, the crack pattern was fully developed; no new cracks formed, and existing cracks just became longer and widened until eventual failure. At this stage, all the deflection monitoring devices were removed to avoid damage. The failure of the beams was brittle, sudden, and accompanied by a loud noise. The crack leading to the failure was formed at the shear span, see Figures 7 and 8. In mid-span, there was no sign of yielding of the tension rebar and no sign of concrete compression failure at the compressive zone of the beam. Clearly, the beam failed due to shear action. Careful examination of the rebar after the test of series II beams indicated that there was no corrosion of rebar after 28 days of sea exposure. Overall, there was no significant difference between the two series in terms of the crack pattern and development.

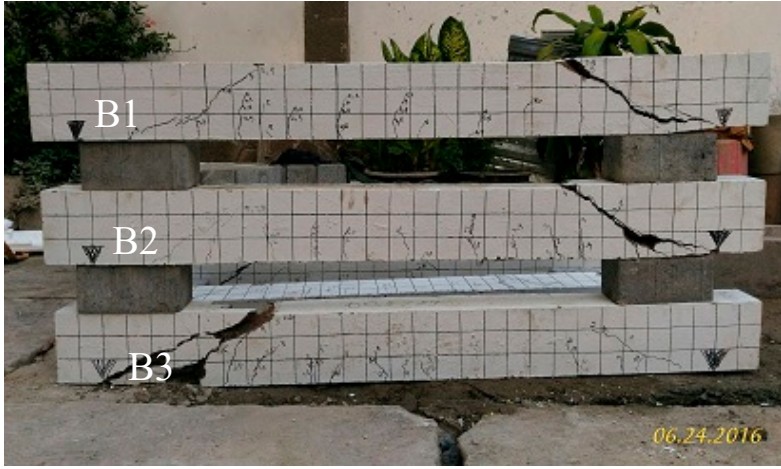

**Figure 7.** Crack pattern of the series I test (B1, B2, and B3).

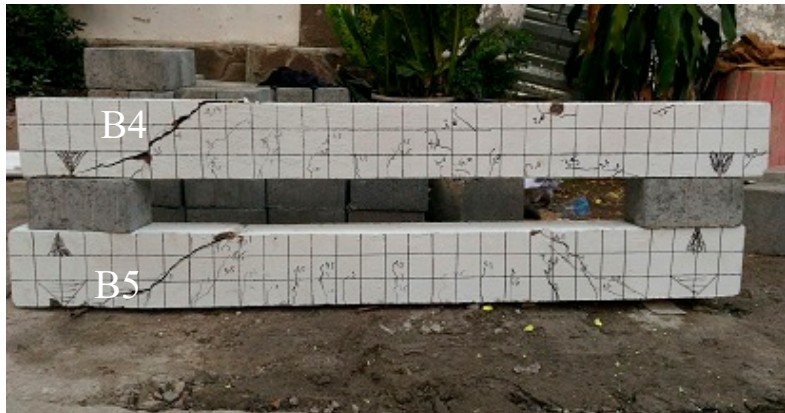

**Figure 8.** Crack pattern of the series II test (B4 and B5).

*3.6. Load Deflection Characteristic*

The load versus mid-span deflection curves until 70% of the ultimate load are shown in Figure 9. In general, the two series followed the same trend even though the compression strength of the first series was 40% higher than that of the compressive strength of the second series. Surprisingly, specimens from series II test which had low compressive strength showed slightly higher stiffness than that of the series I test. Figure 10 shows the stiffness degradation curve for all beams. During loading tests, all the beams followed almost similar strength degradation curves. However, the series II beams, and especially B5 had a higher stiffness than the series I beams. Note that the strength degradation curve only captured up to 70% of the failure load to avoid damage to the deflection monitoring devices.

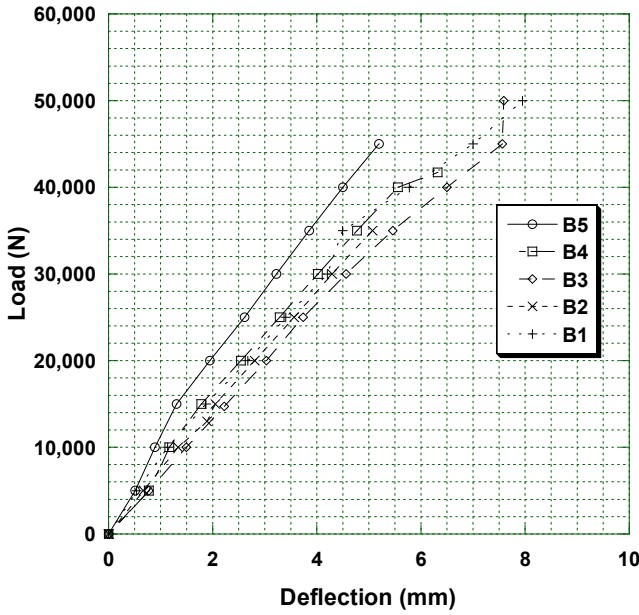

**Figure 9.** The load and deflection curves at mid-span for series I and II tests.

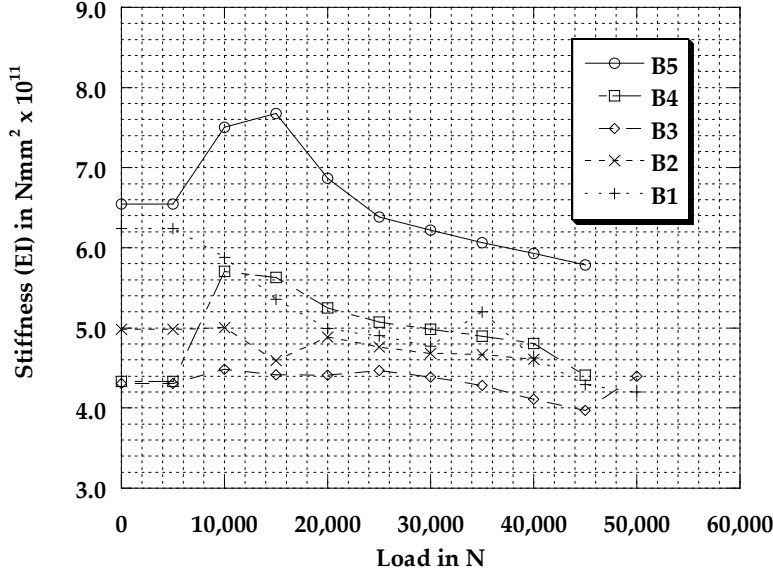

**Figure 10.** The stiffness degradation curve for series I and II tests.

### 3.7. Cracking and Ultimate Load

Table 8 summarizes the cracking and ultimate load of series I and II tests. The cracking load is defined as the load at first crack formation whereas the ultimate load is defined as the load at failure. The P design is defined as the ultimate load of the GCB determined using the American Concrete Institute (ACI) formula for shear strength capacity of normal concrete [31]. The shear strength ($V_n$) in N based on ACI is given as

$$V_n = V_c + V_s \tag{1}$$

$$V_c = \frac{1}{6}\sqrt{f'_c}\, b_w d \tag{2}$$

$$V_s = \frac{A_v f_y d}{s} \tag{3}$$

where:

- $V_c$ = concrete shear strength (N)
- $V_s$ = steel shear reinforcement shear strength (N)
- $f_c$ = concrete strength (MPa)
- $b_w$ = width of beam (mm)
- $d$ = effective depth of beam (mm)
- $A_v$ = area of shear reinforcement (mm$^2$)
- $f_y$ = yield strength of steel reinforcement (MPa)
- $s$ = spacing of shear reinforcement (mm)

For the series I test, the cracking load was around 24%–47% of their ultimate load whereas for the series II test the cracking load was around 60%–65% of their ultimate load. However, due to the limited number of beams tested, it cannot be concluded that the sea environment increases the shear cracking load of geopolymer concrete. Further tests are needed to confirm this, despite some evidence to support this [9]. The shear strength of the GCB generally exceeded the predicted shear strength based on the ACI Code, except for specimen B4. The ultimate load of specimen B4 was only 82% of the ultimate load based on the ACI Code.

**Table 8.** The cracking and ultimate load of series I and II tests.

| Specimen | Cracking Load $P_{cr}$ (N) | Ultimate Load $P_u$ (N) | $P_{cr}/P_u$ | $P_u/P$ design |
|---|---|---|---|---|
| B1 | 25,000 | 74,000 | 0.34 | 1.45 |
| B2 | 13,000 | 53,500 | 0.24 | 1.05 |
| B3 | 32,000 | 68,800 | 0.47 | 1.35 |
| B4 | 25,000 | 41,700 | 0.60 | 0.82 |
| B5 | 46,000 | 71,000 | 0.65 | 1.39 |

## 4. Conclusions and Future Work

This paper considers the shear strength of geopolymer concrete beams using high calcium content fly ash cured in a room and marine environment for 28 days, without high heat curing. From concrete cylinder tests, specimens cured at room temperature of 34 °C had 40% higher average compressive strength than that of specimens cured in seawater environment 25 °C. Specimens cured at room temperature also had slightly lower porosity and high concrete resistivity. The loading test of the GCB until failure showed that the crack pattern and crack development for the two series had no significant differences. However, specimens cured in seawater showed a higher proportion of the cracking load to ultimate load than the specimens cured at room temperature. Further tests are required to understand the effect of the marine environment on the cracking properties of geopolymer concrete to get a convincing conclusion. The shear strength of the GCB generally exceeded the predicted shear strength based on the ACI Code. Future work will focus on formulating mixtures for geopolymer concretes to achieve design compressive strengths between 30–40 MPa without heat curing, extending the setting time of type C FA-based geopolymer concretes, and performing much longer-term durability tests in aggressive environments.

**Author Contributions:** M.S.D. designed the experiment, performed the tests, analyzed test data, and wrote the paper. R.B. performed the tests, analyzed test data, and wrote the paper. H.S. performed the tests and reviewed the draft paper. N.A.H. performed the tests and analyzed test data. R.B.A.A. performed the tests and analyzed test data.

**Funding:** This research was funded by Kemenristekdikti Indonesia (Ministry of Research, Technology, and Higher Education of the Republic of Indonesia).

**Acknowledgments:** The authors thank the Ministry of Research, Technology and Higher Education of the Republic of Indonesia for funding this research.

**Conflicts of Interest:** The authors declare no conflict of interest.

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
