# Peer review of "Shear Strength of Geopolymer Concrete Beams Using High Calcium Content Fly Ash in a Marine Environment"

_buildings, doi:10.3390/buildings9040098_

Reviewer 1 Report

This paper presents an experimental investigation of the shear strength of geopolymer concrete beams using high calcium content fly ash in the marine environment. The following comments are suggested:

1. The authors should cite more recent publications in their literature review.

2. The reaction mechanism of geopolymer concrete has not been consistently recognized. This is because on the one hand, the current diversity of cementitious materials creates the complexity of the reaction mechanism research. On the other hand, the variability of chemical activators exacerbates the uncertainty of the reaction process. Therefore, this paper should emphasize that the research path of the reaction mechanism of geopolymer concrete still needs to be continuously developed and innovated.

3. The shear strength prediction formula of the ACI Code should be listed. However, it must be emphasized that it is only applicable to general cement concrete.

4. How is the spacing of the shear rebar of the beam members determined? It will dominate the failure mode of the beam.

5. Please explain how the beam stiffness is calculated in Figure 10.

6. In lines 206-207: “Table 5 indicates that geopolymer concrete cured in room temperature has a slightly lower porosity that that of geopolymer concrete cured at sea water environment.” The first "that" in the sentence should be "than".

7. The results of each test showed that the variation within the group of the specimens was quite large. The author should clearly explain.

Author Response

1. Done. Additional reference has been added

2. The main purpose of the paper is to investigate the behavior of geopolymer beams under shear load and not the reaction mechanism of geopolymer concrete

3. Done. ACI shear strength formulae has been added

4. The ultimate load of the beam under shear failure is made less than the ultimate load under  flexure, by providing less shear reinforcement that is required for flexural failure to occur.

5. By using deflection formulae and knowing the load and deflection from the test load

6. Done

7. Done.

Reviewer 2 Report

This work presents a study on the mechanical properties including shear strength, crack patterns of geopolymer concrete containing high content of fly ash. 2 curing conditions were tested including room and marine conditions, which leads to a conclusion that room curing offers better mechanical properties and electrical resistivity, while crack patterns and propagation seem not to be affected by curing conditions. Though a number of useful experimental data were shown, still a lot of room to improve the manuscript before any consideration for publication. Revised version should address the following issues:

More extended literature review on geopolymer with respect to mechanical properties, durability is needed in the Introduction.

Line 34: process à offers

Lines 46-48: this should not be the reason for the failure of a scientific research, and should not be based for the current research, which decided that heat curing was not selected,

If table 1 is shown, the authors should compare the results of this study and previous one.

Lines 87-88: aggregate à aggregates; was à were

Table 2: Na2SiO3 has not yet discussed in the text.

Line 101: sentence is not clear, check it again.

Fig. 2: should not show the slump test if you do not have  better photo which captures full geopolymer concrete spreading during slump test.

Lines 128-130: the chemical properties of sea water should be determined. Flow rate is also important to know. These properties would help to explain the results of concrete cured in sea conditions.

How did you make sure that there is no mechanical damage when you transport the beam to the port?

Line 154: is this splitting tensile or flexural tensile strength?

Lines 179-180: BS 1881 does not apply for geopolymer concrete?

Table 4 and the rest: kg is not the unit of force, it should be in N or kG.

Lines 190-192: it is not clear how did you measure electrical resistivity, whether on cubic samples of reinforced beams?

Lines 204-205: it’s not clear what porosity you got from this method, whether total or capillary porosity?

Line 208: how does porosity confirm the mechanical properties?

Table 5: I consider you got the same porosity considering the derivation of the measurement.

The mechanism of degradation/polymerization in marine environment should be discussed.  

Author Response

1. Done, Additional reference has been added

2. Done

3. Agree

4. Previous study dealt with flexure, whereas present study deal with shear. The two studies can not be compared.

5. Done

6. Done

7. Done

8. Figure 2 is still OK

9. Done (new Table for sea water composition has been added)

10. Transportation of the beams using manpower

11. Splitting tensile test using cylinder sample not flexural tensile strength test

12. Agree

13. Done

14. resistivity is measured on concrete beams and not on concrete cylinders

15. Total porosity

16. The measured total porosity did not correlate well with the mechanical properties, especially concrete strength

17. Agree. Only slight different of porosity between the two series

18. This study is only preliminary study and focus on shear strength of concrete beam and used very short time of sea water immersion. To study the mechanism of degradation/polymerization in marine environment required longer time of sea water immersion.

Round  2

Reviewer 2 Report

The revised version is significantly improved, most of reviewers' comments have been addressed. Minor language check should be done before publication.

Buildings EISSN 2075-5309 Published by MDPI AG, Basel, Switzerland RSS E-Mail Table of Contents Alert
Back to Top